# Experimental Study on the Effect of Freeze-Thaw Cycles on the Mechanical and Permeability Characteristics of Coal

Heng Gao [1,2], Jun Lu [1,*], Zetian Zhang [2], Cong Li [2] and Yihang Li [2]

1   Institute of Deep Earth Science and Green Energy, College of Civil and Transportation Engineering,
    Shenzhen University, Shenzhen 518060, China; gaoheng@stu.scu.edu.cn
2   State Key Laboratory of Intelligent Construction and Healthy Operation and Maintenance of Deep
    Underground Engineering, Sichuan University, Chengdu 610065, China; zhangzetian@scu.edu.cn (Z.Z.)
*   Correspondence: junlu@szu.edu.cn

**Abstract:** The safe and efficient mining of coal seams with low porosity, low permeability, and high heterogeneity under complex geological conditions is a major challenge, with the permeability of coal seams playing a crucial role in coal mine gas extraction. The development of coal seam permeability enhancement technology can help coal mines produce safely and efficiently, while the extracted coal bed methane can be utilized as green energy. To study the effect of freezing and thawing on the evolution of the mechanical and permeability properties of coal, triaxial permeability tests were conducted on low-permeability coal under two different confining pressures. Simultaneously, dry, saturated, and freeze-thaw coal samples were set up for comparison, and the effects of water and freeze-thaw were isolated from each other. The triaxial mechanics and percolation laws of dry, saturated, and freeze-thaw coal rocks were obtained; the results show that saturated coal has the lowest initial permeability, while freeze-thawed coal has the highest initial permeability. Through analyzing the effects produced by water, freezing and thawing on coal specimens, the mechanism of the influence of freeze-thaw on the permeability evolution of coal was revealed. The research results can provide theoretical guidance for the development of gas extraction technology for low-permeability coal seams.

**Keywords:** permeability; freeze-thaw; freeze-swelling force; low-permeability coal; permeability increase mechanism

## 1. Introduction

Energy has been an enduring topic in the ongoing process of social and economic development. Coal, as a primary source of energy, has played an indispensable role from certain perspectives and continues to be a major global energy source [1–4]. As shallow coal resources are depleted, the geological conditions of deep coal seams become more complex. Coal reservoirs at these depths are typically characterized by low porosity, low permeability, and high heterogeneity, posing significant challenges and limitations for coal mining operations [4–9]. Therefore, the disposition of coal mine gas and the physical and mechanical properties of the coal rock mass have been a focus of research attention [10,11]. While coalbed methane can pose significant risks in coal mines, it also presents a promising opportunity as a clean, high-quality, and efficient energy source. The efficient exploitation of coal bed methane is of great significance in addressing both the energy shortage and the threat of mine gas disasters [12–14].

Technology for enhancing reservoir permeability has become a focal point of interest, with numerous scholars conducting research in this area using various technical approaches [15]. With the advancement of coal seam fracturing technology, waterless coal seam fracturing, as an emerging technology, has garnered considerable attention. Injecting liquid nitrogen, an extremely low-temperature medium, into a coal seam produces multiple effects. Firstly, during gasification, it absorbs a substantial amount of heat, leading

to the freezing of liquid water into a solid state. This freezing expansion exerts forces that surpass the strength limit of coal and rock, resulting in the formation of new cracks. Secondly, coal consists of various minerals, and the decrease in temperature generates thermal stress, inducing coal fracturing with the aim of increasing gas permeability. Several studies have demonstrated the significant effects of this technology. For instance, McDaniel et al. conducted work on coalbed methane production and showcased how liquid nitrogen can effectively enhance the permeability of coal seams [16]. Grundmann et al. achieved a remarkable 8% increase in gas production compared to conventional methods by fracturing shale rocks (or samples) with liquid nitrogen [17]. A substantial number of scholars have conducted studies on this subject. Winkler et al. demonstrated that water in the pores, fractures, and joints of rocks plays a dominant role in freeze-thaw cycle damage [18]. Chu et al. conducted a comprehensive study on the evolution of pore structure damage, physical and mechanical properties, and permeability characteristics of coal bodies under varying freezing times and numbers of freeze-thaw cycles using liquid nitrogen. Their research involved a combination of experimental and theoretical analysis, allowing them to reveal the mechanism behind the damaging effects of the liquid nitrogen freeze-thaw cycle on coal from two perspectives: thermal stress and freeze-thaw action [19,20]. Takarli et al. conducted an extensive experimental study on the microstructural changes in granite during freeze-thaw cycles [21]. McDaniel et al. demonstrated that injecting liquid nitrogen into the coalbed methane (CBM) reservoir creates a violent temperature shock effect, leading to physical changes in the fracture wall. This phenomenon prevents hydraulic fractures and thermally induced fractures from completely closing under closure stress. Additionally, it creates thermally induced microfractures orthogonal to the hydraulic fractures [16].

These previous studies have explored the effects of liquid nitrogen on the structural and permeability properties of rocks, yielding valuable conclusions. However, the complexity of the alterations caused by the extremely low temperature of liquid nitrogen makes it challenging to distinguish the specific effects of thermal stress from freezing and swelling forces when applied directly to specimens. To address this issue, we conducted freeze-thaw disposal of coal samples at both −18 °C and room temperature. By doing so, we removed the influence factors introduced by ultra-low temperature liquid nitrogen on coal, allowing us to focus solely on the effects of freezing and swelling forces of water on the mechanics and permeability evolution of coal rocks. This experimental study provides a solid theoretical foundation for understanding the mechanism of reservoir seepage and increasing permeability under multi-field coupling conditions. The findings of this study have significant implications for disaster control in deep coal mining and unconventional gas extraction, ultimately improving recovery and safety.

## 2. Experimental Preparation and Method

### 2.1. Rock Description and Specimen Preparation

The study area of this work is the Baijiao coal mine of the Sichuan Coal Group Furong Company in Yibin City, Gongxian County, Southwest China. The coal is anthracite with a natural density between 1.53 and 1.64 g/cm$^3$, and the mineral composition of the coal samples is mainly kaolinite, quartz, calcite, etc. [10]. The coal samples used in the test were taken from the same mining face. Selected intact large coal blocks with good homogeneity were wrapped and sealed with polyethylene cling film and transported to the indoor laboratory for further processing. The large coal samples were drilled into cylinders of 50 mm diameter with a special sampling machine, and the cylindrical coal was cut into 100 mm lengths with a cutting machine and polished into φ × h = 50 mm × 100 mm standard specimens with section non-parallelism and non-verticality less than 0.02 mm (Figure 1). These standard coal samples were sealed with polyethylene film and stored in a cool place.

Before the test, the standard coal samples were divided into a dry group, a water-saturated group, and an experimental group. The dry group was put in a dryer at 60 °C and weighed at intervals of 6 h until the weight of the specimen no longer changed. The water-saturated group was placed into vacuum bottles with good sealing performance. The bottles were filled with water until the coal samples were completely submerged, and a sealing piston was installed to create a vacuum. The coal specimens were periodically removed and weighed every 6 h until their weight remained constant, indicating that they had reached a water-saturated state.

For the experimental group, water-saturated coal specimens were placed in a temperature-controlled chamber set at −18 °C for a duration of 24 h. Afterward, the frozen coal specimens were thawed in room-temperature water (approximately 18 °C) for 24 h. This freeze-thaw cycle was repeated five times.

Notably, there was no significant difference in the appearance of the rock samples in each group before and after treatment.

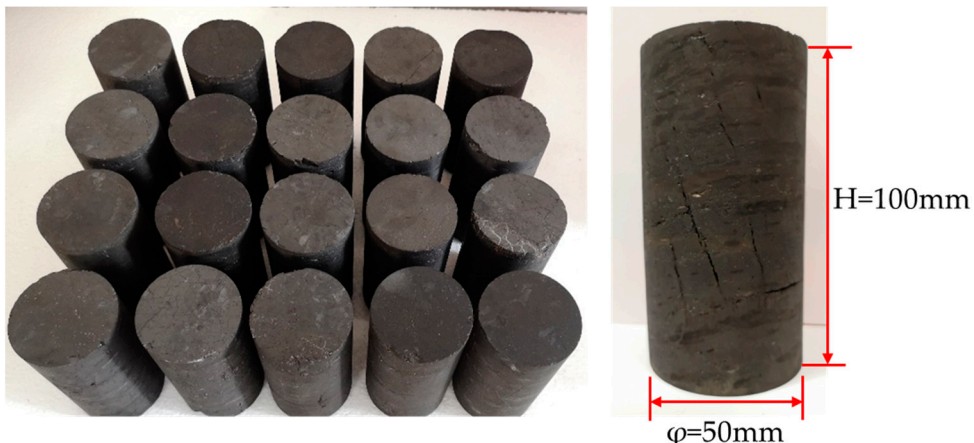

**Figure 1.** Standard coal specimens for testing.

### 2.2. Three-Axis Test Equipment

The study used a fracturing and seepage experimental system for multi-physical field and multiphase coupling of porous media (THM-2) developed by Chongqing University, as shown in Figure 2. The system consists of a main frame, an electrohydraulic servo hydraulic pump station, a pneumatic and hydraulic pressure supply system, and a measurement and control system. The test equipment can provide a maximum axial pressure of 1000 kN and a maximum confining pressure of 60 MPa, and is capable of accepting two specimen sizes ($\varphi$50 mm × 100 mm and $\varphi$100 mm × 200 mm) for the test. The displacement monitoring system can monitor maximum axial and radial displacements of 60 mm and 12 mm, respectively. The system is capable of simulating multiple coupled conditions such as in-situ stress, mining-induced stress, temperature, and fluid pressure, and conducting uniaxial, triaxial mechanical tests, hydraulic fracturing tests, permeability tests, etc., on coal, sandstone, and other materials. It possesses high reliability and accuracy [22,23]. Before conducting the tests, we performed a seal integrity check on the testing equipment to ensure that the permeability system had good sealing performance prior to formal testing.

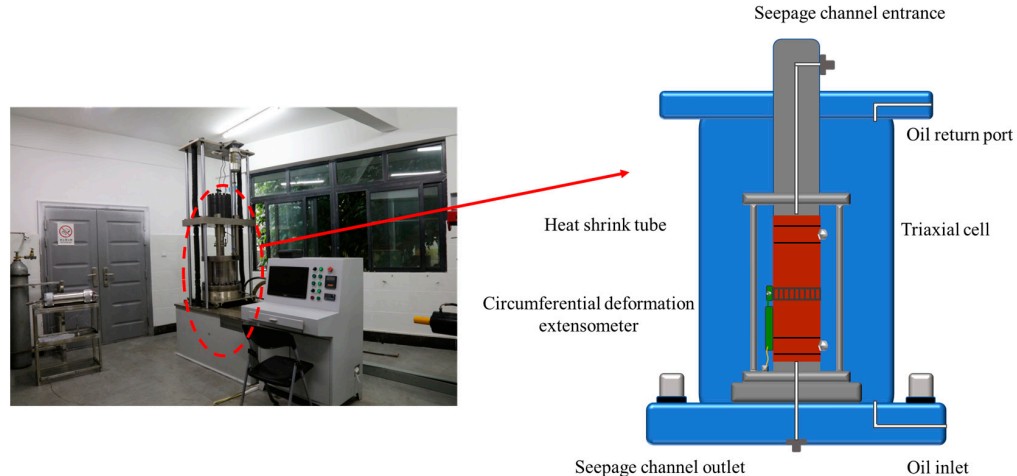

**Figure 2.** Fracturing and seepage experimental system for multi-physical field and multiphase coupling of porous media.

### 2.3. Test Method and Loading Path

To investigate the impact of freeze-thaw cycles on the permeability characteristics of coal, we conducted $CO_2$ seepage experiments on coal specimens in three different states: dry, saturated, and freeze-thawed. These experiments were carried out under confining pressures of 5 MPa and 15 MPa. The testing procedure is presented in Table 1 and is described in detail below.

**Table 1.** Initial stress conditions and test scheme.

| State of Specimen | Specimen Number | The Initial Stress/MPa ($\sigma_1 = \sigma_2 = \sigma_3$) | Confining Pressure ($\sigma_2 \backslash \sigma_3$) | Test Condition | |
| --- | --- | --- | --- | --- | --- |
| | | | | Axial Force | Injection Pressure |
| Dry | CSL5D | 5 | | | |
| | CSL15D | 15 | | | |
| Saturation | CSL5S | 5 | Maintain constant | Displacement loading (0.002 mm/s) | 1.5 MPa gaseous $CO_2$ |
| | CSL15S | 15 | | | |
| Freeze-Thaw | CSL5F | 5 | | | |
| | CSL15F | 15 | | | |

Step 1: Place the standard coal specimen on the lower loading plate (the specimen is concentric with the loading plate). Install O-ring seals on the corresponding pressure rods and loading plates. Use heat-shrinkable tubes to cover the rock and loading plates. Apply heat with a heat gun to ensure a perfect coupling with the O-ring seal and create a pre-seal. Then use clamps for further sealing to prevent any gas leakage during the experiment (silicone gel may be evenly applied on the side of the specimen to form a soft gel layer of about 1 mm to prevent damage to the heat-shrinkable tubes from rocks with large surface pores). Install a radial displacement sensor in the middle of the specimen and ensure it is in good working condition. Use a sample car to transport the prepared specimen into the top part of the triaxial pressure chamber. Lower the triaxial pressure chamber and install the sealing screws of the triaxial pressure chamber.

Step 2: Apply a preload of 0.5 kN axial stress on the specimen as a pre-stress to stabilize the rock sample. Then fill the triaxial pressure chamber with hydraulic oil and close the oil inlet valve. Load the axial stress to 10 kN (30 kN) at a rate of 0.1 kN/s while increasing the confining pressure inside the triaxial chamber to 5 MPa (15 MPa) at a rate of 0.05 MPa/s. Maintain stress stability.

Step 3: Inject $CO_2$ into the specimen from one end along the axial direction at a pressure of 1.5 MPa. Attach a pressure gauge to the other end of the specimen and wait until the pressure reading reaches 1.5 MPa before proceeding to the next step.

Step 4: Replace the pressure gauge at the outlet end with a high-precision flow meter. Keep the confining and axial pressures constant. Monitor the flow meter reading until it stabilizes and then start recording data. Meanwhile, load axial stress at a rate of 0.002 mm/s until fracture.

## 3. Experimental Results and Analysis

### 3.1. Mechanical Response Characteristics of the Coal in Different States and Confining Pressures

Figure 3 displays the stress–strain curves of the coal specimens in three different states under two distinct confining pressures. The stress–strain characteristics of the coal samples align with the general rock mechanics testing law, exhibiting four stages during the triaxial compression process: compression densification (which may not be apparent under high confining pressure), elastic deformation, plastic rupture, and fracture. It was observed that the triaxial compressive strength and residual strength of the specimens under low confining pressure were significantly higher than those under high confining pressure. As the axial stress increases, positive strains are generated in the axial direction, while negative strains are generated in the radial direction. Following the fracture of the specimens, there is a subsequent process of volume increase (expansion).

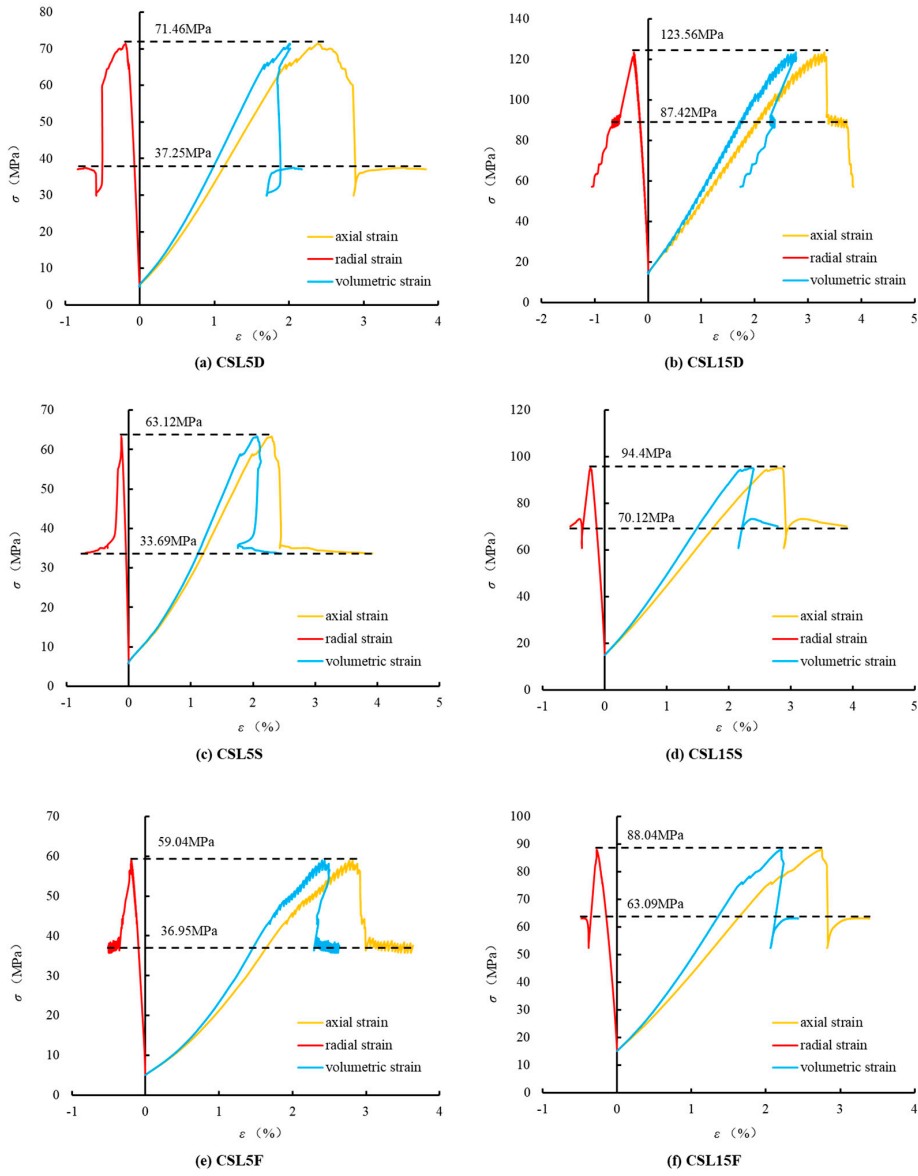

**Figure 3.** Stress–strain curves of coal under different test conditions.

Comparing the test results under the two different confining pressure conditions, it was found that the dry coal specimens exhibited higher strength and the freeze-thawed coal specimens had lower strength. Under the 5 MPa confining pressure condition, the strength of the saturated specimen was 63.12 MPa, which was 11.67% lower than the strength of the dry coal. The strength of the freeze-thawed specimen was 59.04 MPa, indicating a 17.38% decrease in strength compared to the dry coal. Similarly, under the 15 MPa confining pressure condition, the strength of the saturated specimen was 94.4 MPa, reflecting a 23.6% reduction compared to that of the dry coal. The freeze-thawed specimen exhibited a strength of 88.04 MPa, which was 28.75% lower than that of the dry coal.

### 3.2. The Permeability Evolution Characteristics of Coal in Different States and Confining Pressures

This section reports the permeability evolution characteristics of the coal in different states under different confining pressures. The flow rate of gas through the specimen is recorded during the test. Since we temporarily disregarded the effect of temperature and the whole experiment was conducted at a constant temperature in this study (the indoor laboratory temperature is about 20 °C), the transport of $CO_2$ inside the coal can be regarded as a constant-temperature laminar flow. This satisfies the basic conditions of Darcy's law [24]. The expression used to calculate the permeability is given by Equation (1):

$$k = \frac{2P_0Q\mu L}{A\left(P_1^2 - P_2^2\right)} \tag{1}$$

where $k$ is the effective permeability (m$^2$), $Q$ is the gas seepage flow (m$^3$/s), $P_0$ is the standard atmospheric pressure of about 0.1 MPa (MPa), $\mu$ is the $CO_2$ gas dynamic viscosity, $1.4932 \times 10^{-11}$ MPa·s (MPa·s), $L$ is the length of the specimen (m), $P_1$ is the inlet gas pressure, 1.5 MPa (MPa), $P_2$ is the outlet gas pressure, 0.1 MPa (MPa), and $A$ is the effective area of gas seepage (m$^2$).

The permeability evolution curves of dry coal under two different confining pressures (5 MPa, 15 MPa) are shown in Figure 4. The permeability of the coal rock specimens under low confining pressure (5 MPa) is approximately 0.2 to $1 \times 10^{-17}$ m$^2$ during the early stage of loading. As the test proceeds, the axial stress gradually increases and several brief increases in permeability (marked by green circles) occur, increasing to 0.6–$2 \times 10^{-17}$ m$^2$. Note that each brief increase in permeability is quickly followed by a decrease in the initial value before the increase. At the peak of the axial stress, the specimen breaks down and the permeability suddenly increases steeply (marked by the yellow rectangle), at which point the magnitude of the increase exceeds that of any previous increase (by an order of magnitude, with a maximum value of $14.03 \times 10^{-17}$ m$^2$), and as the test proceeds, the permeability quickly returns to its initial level at the post-peak stage of the test. Under high confining pressure (15 MPa), the permeability of the coal specimens ranged from roughly 0–$0.23 \times 10^{-17}$ m$^2$. At the start of the loading, the initial permeability experienced a notable decrease compared to the conditions observed in the low-confining-pressure tests. As the test progressed and the axial stress gradually increased, the permeability remained relatively stable without significant changes. However, when the axial stress reached its peak, just before specimen fracture, the permeability exhibited a sudden steep increase (highlighted by the yellow rectangle) to a value of $44.81 \times 10^{-17}$ m$^2$. Following this sharp increase, the permeability quickly returned to the magnitude observed at the beginning of the test.

Numerous studies have consistently demonstrated that triaxial permeability tests lead to a decrease in rock permeability during the initial stage, owing to the compression of pores and cracks. As the test progresses into the crack extension phase, the permeability response shows an increase, attributed to the enlargement of pores and cracks resulting from increased rock damage. It is important to emphasize that this damage is an irreversible process. However, the above test results yield a different conclusion. Despite the expected decrease in permeability during the initial stage, the permeability suddenly and significantly increases just before the specimen fractures, returning to its initial level

thereafter. This unexpected behavior may be influenced by specific factors or mechanisms inherent to the conditions or material properties of the tested coal specimens. A more comprehensive investigation and analysis are warranted to fully understand and interpret the observed permeability response in this study.

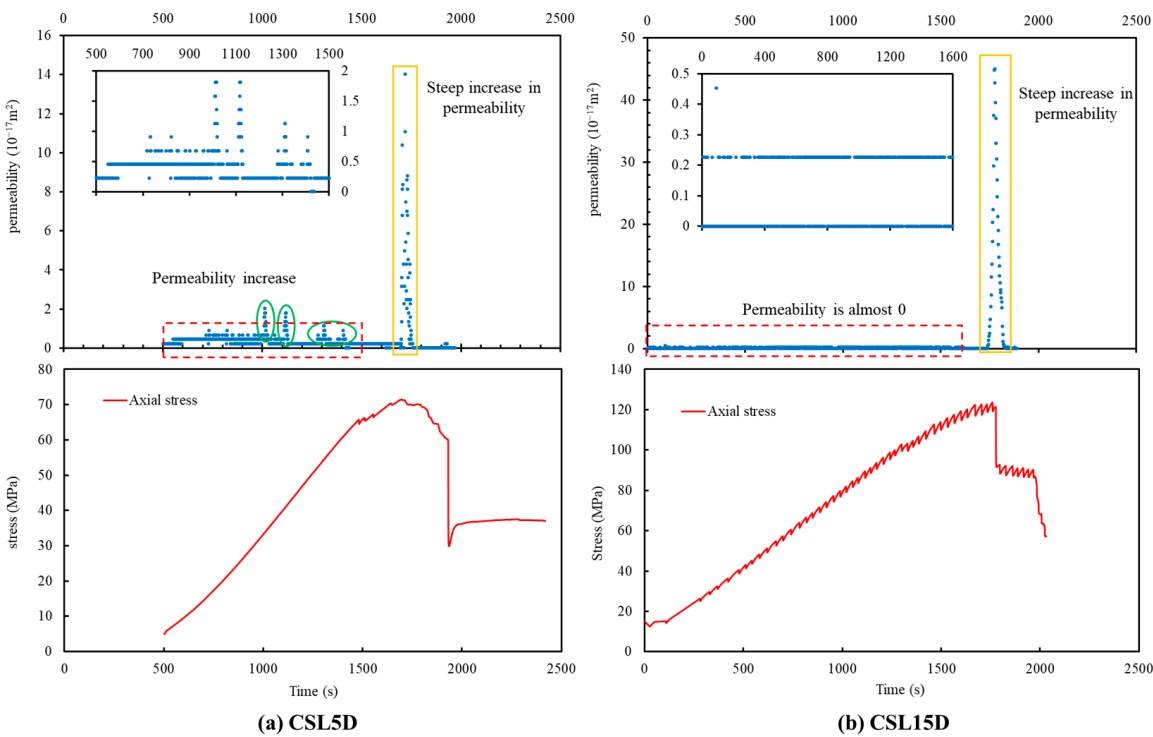

**Figure 4.** Permeability evolution of dry coal under different test conditions.

The permeability evolution curves of saturated coal under two confining pressures (5 MPa, 15 MPa) are presented in Figure 5. Under low confining pressure (5 MPa), the permeability of the coal specimens starts at 0 $m^2$ during the early loading stage. As the test progresses, the axial stress gradually increases, and the permeability of the coal rock specimen remains mostly at 0 $m^2$, with a few instances at $0.23 \times 10^{-17}$ $m^2$. When the axial stress reaches its peak, the specimen fractures, and the permeability gradually increases to $6.56 \times 10^{-17}$ $m^2$, followed by a gradual decrease. Throughout the test, there is an initial period of permeability increase followed by a steady decrease in the post-peak phase, during which the residual stress after the peak remains essentially constant.

Under high confining pressure (15 MPa), the permeability of the coal specimen remains stabilized at 0 $m^2$ from the start of loading. Compared to the low-confining-pressure test conditions where a small amount of gas seepage is present, the seepage channel is completely blocked at the early loading stage. As the test progresses and the axial stress gradually increases, the permeability does not show significant improvement. However, just before reaching the peak axial stress when the specimen fractures, the permeability experiences a steady increase, accelerating to $3.4 \times 10^{-17}$ $m^2$, and then quickly stabilizes at $1.81 \times 10^{-17}$ $m^2$.

Comparing the test results of dry coal with those of saturated coal, it is evident that the permeability of saturated coal is significantly better under the same stress conditions. However, when the coal is subjected to the influence of pore water and stress, the seepage channels within the coal are completely closed, resulting in almost zero permeability. This issue is prevalent in many low-permeability coal seams, leading to challenges in coalbed methane extraction and utilization, as well as difficulties in managing mine gas disasters. Furthermore, the permeability of saturated coal is notably lower when the coal rock specimen is damaged due to compression. This observation suggests that the presence

of pore water not only affects the permeability of natural pore throats, microfractures, and joints within coal seams, but also impacts the permeability of artificially generated macrocracks. This limitation extends to hydraulic fracturing techniques and other methods employed to fracture coal seams in order to increase permeability.

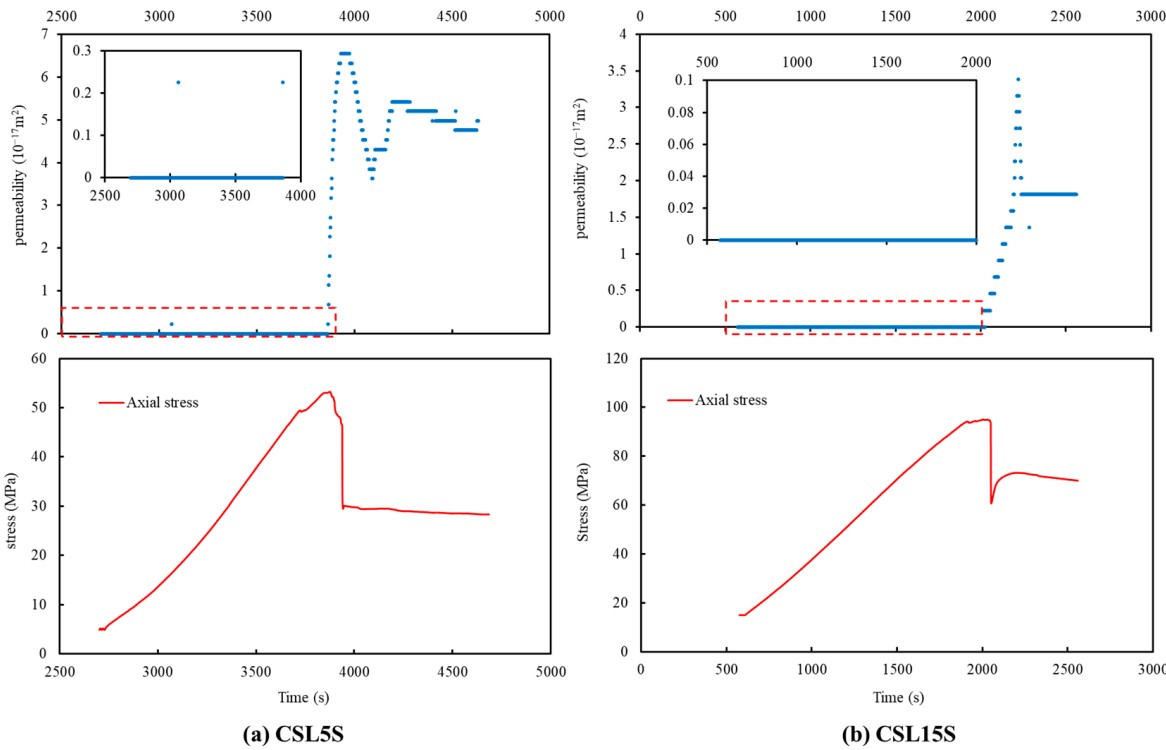

**(a) CSL5S**　　　　　　　　　　　　　**(b) CSL15S**

**Figure 5.** Permeability evolution of saturated coal under different test conditions.

The presence of pore water, therefore, has significant implications for the overall permeability of coal and presents challenges for various coal exploitation and extraction methods, calling for more comprehensive strategies to address and manage permeability issues in low-permeability coal seams.

The permeability evolution curves of the coal after freeze-thawing at two confining pressures (5 MPa, 15 MPa) are shown in Figure 6. The permeability of the coal specimens under the low confining pressure (5 MPa) is $14.68 \times 10^{-17}$ m$^2$ at the early stage of loading. As the test proceeds, the axial stress gradually increases and the permeability of the coal specimen gradually decreases to 0. At the peak of the axial stress, the specimen fractures and the permeability gradually increases to $12.24 \times 10^{-17}$ m$^2$, and then gradually decreases. As the test proceeds (in the post-peak phase), the permeability is consistent with the characteristics of saturated coal, with a period of increase followed by a steady decrease, during which the post-peak residual stress remains essentially constant. Under high confining pressure (15 MPa), the permeability of the coal specimens is stabilized at 0 at the early stage of loading, indicating that the seepage channels generated by freeze-thaw action are completely blocked under the stress. As the test proceeds, the axial stress gradually increases and the permeability does not improve significantly. Just before the axial stress reaches the peak specimen fracture, the permeability first increases steadily and then accelerates to $5.88 \times 10^{-17}$ m$^2$, and finally decreases gradually to a stable value of $3.85 \times 10^{-17}$ m$^2$.

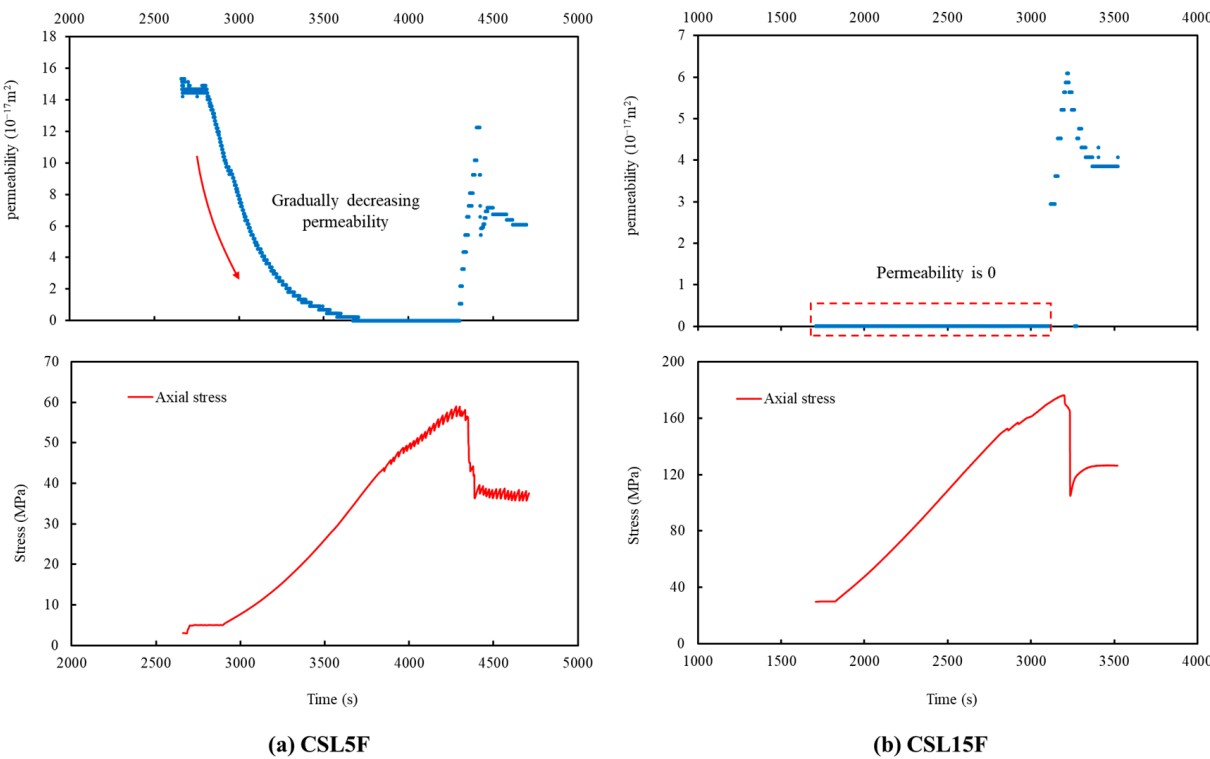

**Figure 6.** Permeability evolution of freeze-thaw coal under different test conditions.

## 4. Discussion

In order to discuss the evolution of permeability in coal specimens in different states (dry, saturated, freeze-thaw) and to reveal the mechanism of freeze-thaw damage on the permeability evolution of coal and the gas migration within coal when stress changes occur during mining activities, this section of the article focuses on the theoretical evolution of coal deformation and the macroscopic permeability of the specimens to explain the observed experimental phenomena.

### 4.1. Analysis of Strain Variation Rate and Stress Variation Rate of Coal

As shown in Figure 7, the curves illustrate the relationship between radial strain rate, axial stress variation rate, and time for coal specimens in different states tested under low confining pressure (5 MPa). The stress variation rates for dry coal and water-saturated coal show relatively consistent behavior. During constant-axial-displacement loading testing, the stress variation rate initially increases, then decreases, ultimately reaching a negative value at the point of fracture and remaining relatively stable after the peak. However, for freeze-thawed coal, the stress rate becomes negative before fracture, with the smallest variation rate observed during the pre-peak stage. This indicates that the combined effects of thermal stress and freeze expansion force induce significant damage in the coal. Additionally, the action of the expansion force results in the enlargement of pore throats, contributing to improved plastic behavior of the freeze-thawed coal.

The radial strain variation rate reflects the deformation, pore deformation, and cracks caused by the Poisson effect. During the initial loading stage, all three states of coal exhibit a stable and relatively small radial strain variation rate (expansion). At this stage, the primary factors influencing the behavior are the Poisson's ratio of the coal specimens and the mechanical properties of the pore throats. As the testing progresses, the freeze-thawed coal demonstrates accelerated expansion characteristics first, followed by the dry coal. This is attributed to the cumulative effect of cracks. Eventually, when macroscopic cracks form, the radial strain variation rate of the specimens in all three states increases significantly.

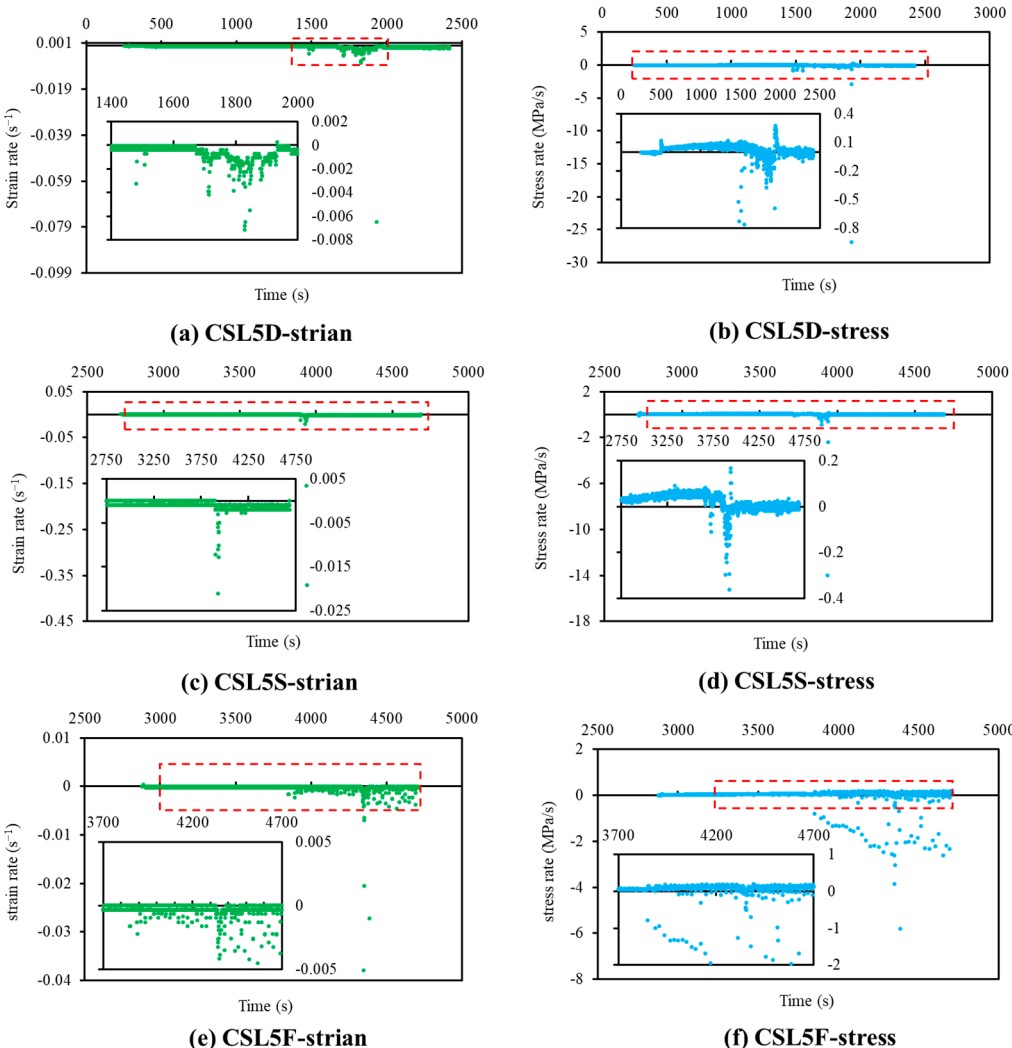

**Figure 7.** Strain variation rate and stress variation rate of coal under low confining stress (5 MPa).

These findings indicate that the presence of water-saturated and freeze-thaw conditions both have an impact on the physical structure and mechanical properties of coal pores. The presence of pore water inhibits radial deformation of the rock and reduces the bearing capacity of the pores, and thereby affects the axial load-bearing capacity of the rock. On the other hand, freeze-thaw actions promote radial deformation of the rock.

The curves of radial strain and axial stress variation rate versus time for coal specimens under different conditions of high confining pressure (15 MPa) are shown in Figure 8. The stress variation rate of coal in the three states gradually increases at the beginning of loading and starts to show negative values as the test proceeds, which means that the stress appears to decrease under the condition of axial constant displacement loading, indicating that damage to the specimen has occurred. This phenomenon appears earliest in dry coal, and in freeze-thaw coal when it undergoes plastic deformation before fracture. This indirectly confirms that freeze-thaw coal suffers additional damage before loading.

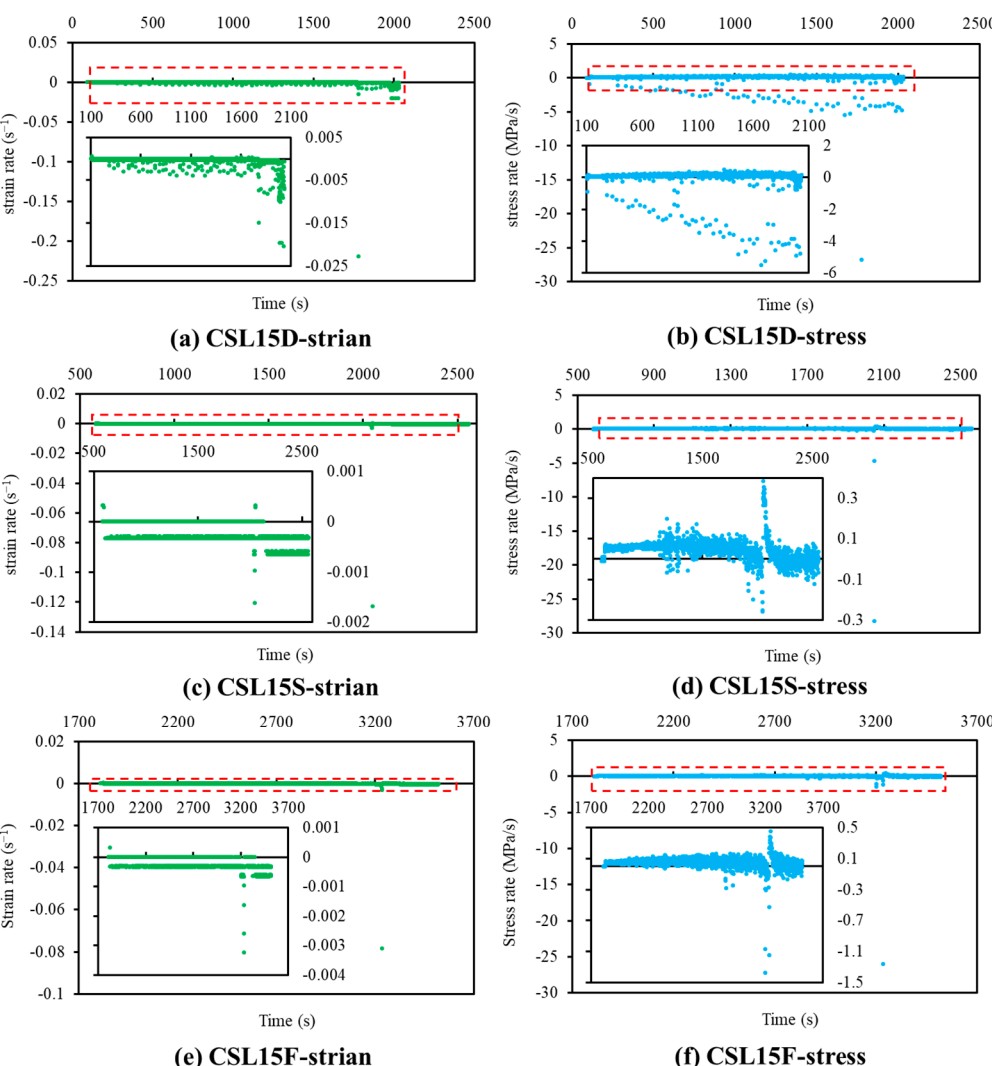

**Figure 8.** Strain variation rate and stress variation rate of coal under high confining pressure (15 MPa).

The rate of radial strain variation is much greater in dry coal than in coal in other states, and the presence of water reduces the Poisson's ratio of the coal while the freeze-thaw action seems to increase the Poisson's ratio of the coal, but this effect is still smaller under high confining pressure conditions. The overall trend of strain variation rate for dry coal continued to decrease throughout the test, which indicates that radial expansion is increasing. In contrast, saturated and freeze-thaw coal expand to a maximum at the moment of fracture and then return to their original rates.

### 4.2. Analysis of the Mechanism of Coal Permeability Evolution

Based on the above experimental phenomena, it is essential to discuss the factors that influence the deformation and evolution of permeability in coal specimens. Coal formations possess natural pores that develop over long-term diagenesis and geological processes, but these pores are often inadequate to meet gas migration requirements during production. As the specimens are subjected to stress, these pores gradually close (referred to as the pore crack compaction stage during the initial loading), resulting in reduced permeability and transitioning into the elastic stage. The variations in pore structure and state of the specimens also contribute to their distinct performances.



When there is no water in the coal pores, connected pores consisting of pores, throat channels, fractures, joints, cleavage planes, and bedding planes serve as gas flow channels. Disconnected closed pores and semi-closed pores become potential gas flow channels, resulting in relatively higher permeability of the coal specimen. As illustrated in Figure 9, under the influence of stress, the flow channels gradually narrow or even close, leading to reduced permeability or even loss of gas permeability. When the stress further increases, microcracks gradually initiate, and at this point, the permeability will increase. As the testing progresses, the damage intensifies, and the detachment of rock particles around the pores causes blockages in the flow channels, resulting in a reduction in permeability. Eventually, when the specimen fractures and macroscopic cracks form, the experimental results demonstrate that under the influence of stress, these macroscopic cracks quickly become blocked.

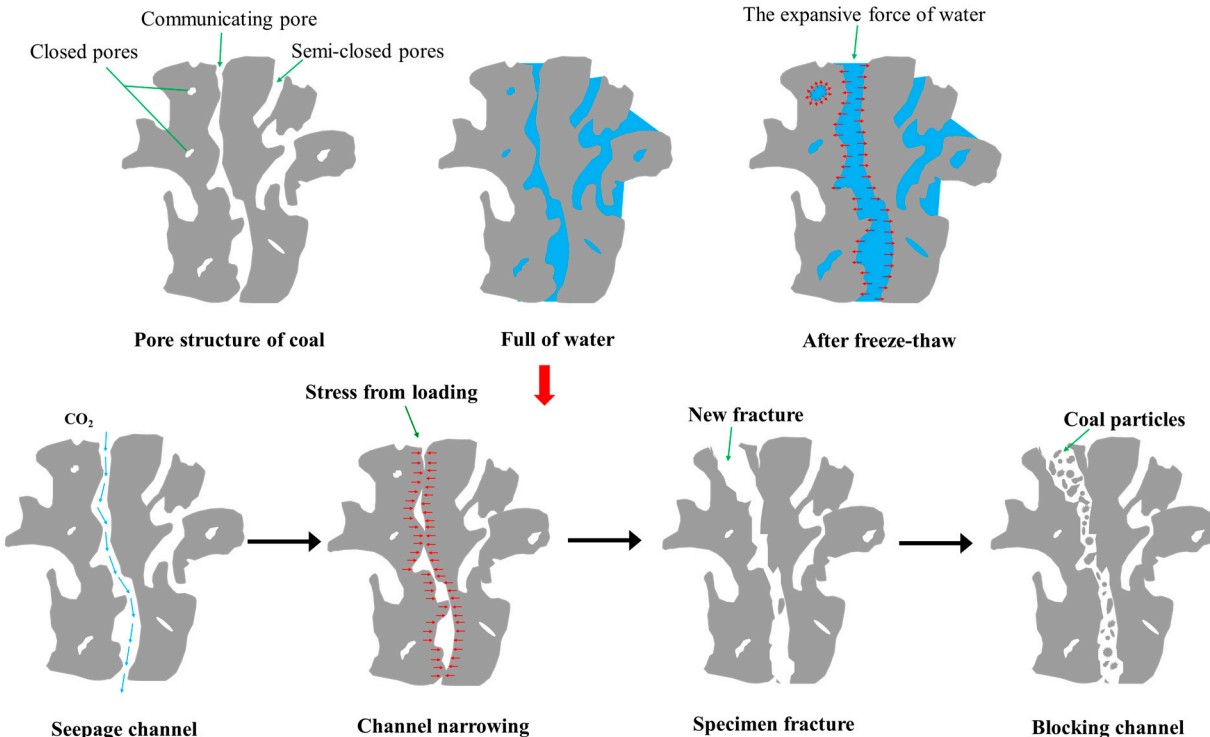

**Figure 9.** Evolution of pore throats and cracks.

When the coal is saturated with water, the pore space inside the specimen becomes occupied by water, obstructing the passages originally intended for gas flow and further reducing the permeability of the coal. Under the influence of stress, the pore water pressure increases, making it more challenging for the gas to pass through. However, when the specimen fractures to produce macroscopic cracks, the presence of pore water can effectively prevent coal particles from blocking the channels, allowing for better gas flow. Freeze-thaw coal exhibits larger pore fissure spaces, making it more conducive to the formation of gas flow channels. During the freeze-thaw process, water undergoes expansion and contraction, which may lead to the destruction of coal samples, enlargement of pores, and increased connectivity, ultimately resulting in increased permeability.

Additionally, the mineral composition of coal can also influence its permeability. Different types of mineral components have varying physical and chemical properties that can affect coal samples differently. Some minerals may undergo physical or chemical changes during the saturation and freeze-thaw processes, potentially increasing the permeability of coal samples.

Having many pores and cracks is the key to the permeability of coal [25,26]. The permeability evolution pattern of dry, water-saturated, and freeze-thawed coal in axially loaded tests will be discussed. Therefore, the permeability model shown in Equation (2) is included in the scope of this discussion. This model was proposed by Shi et al. [27] and is known as the S and D model:

$$k = k_0 e^{-3C_f(\sigma_1 - \sigma_{1\text{-}0})} \tag{2}$$

where $C_f$ is the cleat volume compressibility, $\sigma_1$ is the axial stress, and $\sigma_{1\text{-}0}$ is the initial stress.

In Equation (2), the cleat volume compressibility was extended to the fracture compressibility, and the fracture compressibility $C_f$ is described as a constant in this model. Existing studies, however, have found that fracture compressibility is not constant, but changes with varying stress conditions. The permeability of our coal specimens progressed through elastic and plastic stages, with the permeability decreasing and then increasing. The reason for the decrease in permeability in the elastic stage is due to the extrusion of the pore throat, while the reason for the increase in permeability in the plastic stage is due to the expansion of cracks, and the value of $C_f$ is changed [28,29]. In simple terms, fracture compressibility is a function of stress, as shown in Equation (3):

$$\begin{aligned} k &= k_0 e^{-3\overline{C_f}(\sigma_1 - \sigma_{1\text{-}0})} \\ \overline{C_f} &= \frac{C_{f0}}{\alpha(\sigma_1 - \sigma_{1\text{-}0})}\left(1 - e^{-\alpha(\sigma_1 - \sigma)}\right) \end{aligned} \tag{3}$$

where $\overline{C_f}$ is the mean equivalent fracture compressibility, $C_{f0}$ is the initial equivalent fracture compressibility, and $\alpha$ is the declining rate of equivalent fracture compressibility with increasing stress.

Figure 10 shows the evolution of the permeability of frozen-thawed coal with axial stress under 5 MPa confining pressure. The permeability of the coal tends to be close to 0 as the axial pressure gradually increases to 40 MPa. The $R^2$ of 0.9975 is obtained by fitting the results, indicating that the freeze-thaw coal can be well-fitted with the S and D model. During this period of testing, the pores and cracks of the coal involved in the seepage channels, which are either natural or artificial, gradually close to blockage.

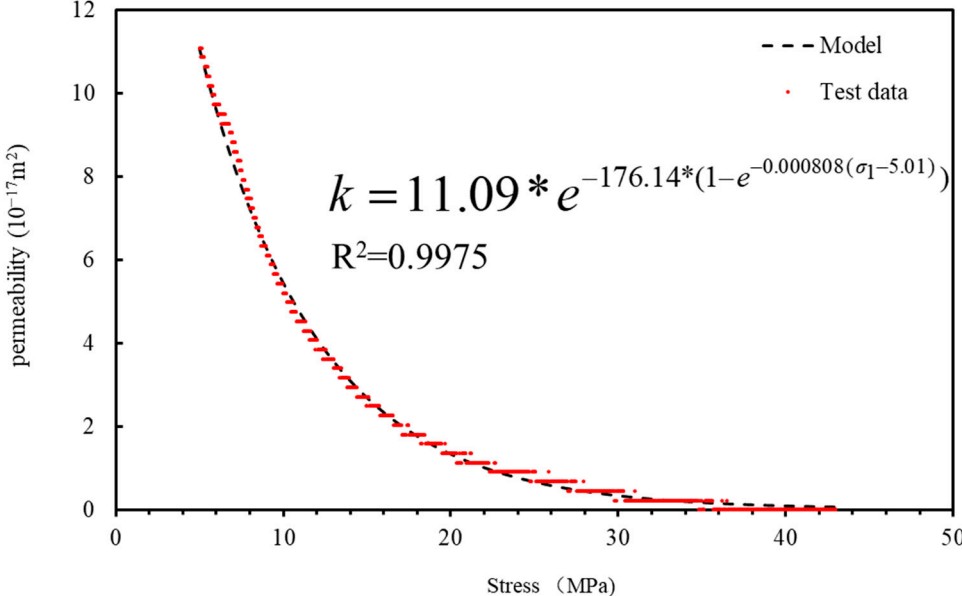

**Figure 10.** Comparison of experimental data of freeze-thaw coal under 5 MPa with the S and D model.

## 5. Conclusions

Self-developed percolation equipment (THM-2) was used for testing, and the mechanical and permeability characteristics of coal in three different states under different confining pressure conditions were studied in order to systematically analyze the influence of freeze-thaw cycles on the mechanical and permeability characteristics of coal. The following conclusions were obtained:

1.  The triaxial compressive strength of dry coal rock specimens is the highest and the triaxial compressive strength of freeze-thaw coal rock specimens is the lowest under the same confining pressure (5 MPa, 15 MPa) and permeability pressure (1.5 MPa $CO_2$).
2.  The cracking and fracture process generates fine coal particles that can obstruct seepage channels, leading to a decrease in permeability. However, pore water can alleviate channel blockage caused by this factor.
3.  The presence of pore water in coal rock specimens will greatly reduce the permeability of coal rock, and freeze-thawing will improve this result to some extent.
4.  The permeability of coal rock after freeze-thaw damage is still consistent with the S and D model.

**Author Contributions:** Software, C.L.; Investigation, H.G.; Resources, Z.Z.; Data curation, Y.L.; Writing—original draft, H.G.; Supervision, J.L.; Funding acquisition, J.L. All authors have read and agreed to the published version of the manuscript.

**Funding:** This research was funded by the National Natural Science Foundation of China (Grant Nos. 52104209 and 52174084), National Key R&D Program of China (No. 2022YFB3706605), Sichuan Science and Technology Program (No.2023NSFSC0919), and Postdoctoral R&D Program of Sichuan University (No. 2023SCU12122).

**Institutional Review Board Statement:** Not applicable.

**Informed Consent Statement:** Not applicable.

**Data Availability Statement:** The data findings and results can be provided upon reasonable request.

**Conflicts of Interest:** The authors declare no conflict of interest.

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
