# Peer review of "Experimental Study on the Effect of Freeze-Thaw Cycles on the Mechanical and Permeability Characteristics of Coal"

_sustainability, doi:10.3390/su151612598_

Round 1
Reviewer 1 Report
The authors have conducted a useful experimental study to understand the effect of freeze-thaw on the mechanical and permeability characteristics of coal.
While the overall study and results are reasonable and useful for the scientific community, the manuscript is full of errors and I strongly suggest that the authors rewrite this document with consultation from a native-English language speaker or the English department in their institute.
- In the first line of abstract: the word "inhomogeneity" should be replaced with "heterogeneity".
- Introduction section, para 2, line 1: "The technology of increase reservoir permeability has been the focus" should be "The technology to increase reservoir permeability has been the focus"
- Introduction section, para 2, line 3 "Liquid ni- trogen is used as a medium to inject into the coal seam, cause the stress difference between minerals, the freezing and swelling force generated by the freezing of liquid water phase change, resulting in new cracks in the coal seam and achieving the purpose of increasing permeability" does not make sense, please rewrite it.
- Introduction section, para 2, line 5 "by fracturing shale logs with liquid nitrogen" should probably be "by fracturing shale rocks (or samples) with liquid nitrogen"
- Introduction section, para 2, line 6 "Chu et al. studied the pore fracture structure damage evolution law" should be "pore structure damage evolution law"
-Introduction section, para 3 "of the water phase change" should be "during the phase change of water"
- Remove word "Therefore" after the above line
- Introduction section, para 3, last line "This is an important guideline for the power disaster control of deep coal min- ing and unconventional gas extraction to improve recovery." What is an important guideline? Maybe replace the word guideline by learning. Typically guidelines are standard procedures in a well-established process that should be replicated to achieve accurate results. Here, the word "learning" seems more appropriate.
- Section 2.1, Para 1, last line "The made standard coal sample is" should be "The standard coal sample is"
- Section 2.1, Para 2 "The water-saturated group is to put the coal sample in the vacuum bottle" does not make any sense. Please rewrite. Do you mean the water saturated coal samples are put in the vacuum bottle"?
- Section 2.1, Para 2 "The experimental group was to place the coal samples completed with full water" does not make sense. Please rewrite.
- Figure 1: Please annotate the diameter and height of the cylinder in the figure itself, so it is easy for the readers to judge the scale of samples.
- Section 2.2, Para 1, line 1 "In this study, fracturing and seepage experimental system for multi-physical field and multiphase coupling of porous media (THM-2) developed by Chongqing University, as shown in Fig. 2." does not make sense. Did you forget to say "was used" at the end?
- Section 2.2, Para 1, line 2 "measurement and con- trol system, etc." please do not write "etc." without giving either a reference to the complete equipment specifications, or writing the actual complete list explicitly.
- Section 2.2, Para 1, line 5 "The system can simulate porous media". Did you mean "The system can test (or perform experiments on) porous media"?
These are just a few errors, but I suggest the authors rewrite the manuscript and carefully proof-read it while consulting a native English language speaker.
While the overall study and results are reasonable and useful for the scientific community, the manuscript is full of errors and I strongly suggest that the authors rewrite this document with consultation from a native-English language speaker or the English department in their institute.
Author Response
Point-by-point Responses to the reviewers’ comments
Reviewer #1:
Comment 1: In the first line of abstract: the word "inhomogeneity" should be replaced with "heterogeneity".
Response:
Thank the reviewers for their professional suggestions. The authors have revised the manuscript for inappropriate expressions.
(In the first line of abstract) The safe and efficient mining of coal seams with low porosity, low permeability and high heterogeneity under complex geological conditions is a major challenge, among which the permeability of coal seams is crucial for coal mine gas extraction.
Thanks very much for your serious attitude and professional advice. All modified parts of the revised manuscript have been marked in red.
Comment 2: Introduction section, para 2, line 1: "The technology of increase reservoir permeability has been the focus" should be "The technology to increase reservoir permeability has been the focus"
Response:
Thank the reviewers for their professional suggestions. The authors have revised the manuscript for inappropriate expressions:
(Introduction section, line 39) The technology to increase reservoir permeability has been the focus of attention, and many scholars have been conducting research in this area using different technical means.
Thanks very much for your serious attitude and professional advice. All modified parts of the revised manuscript have been marked in red.
Comment 3: Introduction section, para 2, line 3 "Liquid nitrogen is used as a medium to inject into the coal seam, cause the stress difference between minerals, the freezing and swelling force generated by the freezing of liquid water phase change, resulting in new cracks in the coal seam and achieving the purpose of increasing permeability" does not make sense, please rewrite it.
Response:
Thank the reviewers for their professional suggestions. The authors have revised the manuscript for inappropriate expressions:
(Introduction section, line 42) When liquid nitrogen, an extremely low-temperature medium, is injected into a coal seam, it yields multiple effects. Firstly, during gasification, it absorbs a substantial amount of heat, causing the freezing of liquid water into a solid state. This creates freezing expansion forces that surpass the strength limit of coal and rock, resulting in the formation of new cracks. Secondly, the coal contains diverse minerals, and the decrease in temperature generates thermal stress, which induces coal fracturing, aiming to increase gas permeability.
Thanks very much for your serious attitude and professional advice. All modified parts of the revised manuscript have been marked in red.
Comment 4: Introduction section, para 2, line 5 "by fracturing shale logs with liquid nitrogen" should probably be "by fracturing shale rocks (or samples) with liquid nitrogen"
Response:
Thank the reviewers for their professional suggestions. The authors have revised the manuscript for inappropriate expressions:
(Introduction section, line 52) Grundmann et al. produced 8% more gas by fracturing shale rocks (or samples) with liquid nitrogen compared to conventional methods.
Thanks very much for your serious attitude and professional advice. All modified parts of the revised manuscript have been marked in red.
Comment 5: Introduction section, para 2, line 6 "Chu et al. studied the pore fracture structure damage evolution law" should be "pore structure damage evolution law"
Response:
Thank the reviewers for their professional suggestions. The authors have revised the manuscript for inappropriate expressions:
(Introduction section, line 55) Chu et al. studied the pore structure damage evolution law
Thanks very much for your serious attitude and professional advice. All modified parts of the revised manuscript have been marked in red.
Comment 6: Introduction section, para 3 "of the water phase change" should be "during the phase change of water"
Response:
Thank the reviewers for their professional suggestions. The authors have revised the manuscript for inappropriate expressions:
(Introduction section, line 68) However, the extremely low temperature of liquid nitrogen generates numerous intricate alterations when directly applied to specimens, rendering the differentiation of effects caused by thermal stress and freezing and swelling forces unfeasible.
Thanks very much for your serious attitude and professional advice. All modified parts of the revised manuscript have been marked in red.
Comment 7: Remove word "Therefore" after the above line
Response:
Thank the reviewers for their professional suggestions. The authors have revised the manuscript for inappropriate expressions:
The authors have removed the word "Therefore".
Thanks very much for your serious attitude and professional advice.
Comment 8: Introduction section, para 3, last line "This is an important guideline for the power disaster control of deep coal mining and unconventional gas extraction to improve recovery." What is an important guideline? Maybe replace the word guideline by learning. Typically guidelines are standard procedures in a well-established process that should be replicated to achieve accurate results. Here, the word "learning" seems more appropriate.
Response:
Thank the reviewers for their professional suggestions. The authors have revised the manuscript for inappropriate expressions:
(Introduction section, line 77) This is an important learning for the power disaster control of deep coal mining and unconventional gas extraction to improve recovery.
Thanks very much for your serious attitude and professional advice. All modified parts of the revised manuscript have been marked in red.
Comment 9: Section 2.1, Para 1, last line "The made standard coal sample is" should be "The standard coal sample is"
Response:
Thank the reviewers for their professional suggestions. The authors have revised the manuscript for inappropriate expressions:
(Section 2.1, line 91) The standard coal sample is sealed with polyethylene film and stored in a cool place.
Thanks very much for your serious attitude and professional advice. All modified parts of the revised manuscript have been marked in red.
Comment 10: Section 2.1, Para 2 "The water-saturated group is to put the coal sample in the vacuum bottle" does not make any sense. Please rewrite. Do you mean the water saturated coal samples are put in the vacuum bottle"?
Response:
Thank the reviewers for their professional suggestions. The authors have revised the manuscript for inappropriate expressions:
(Section 2.1, line 96) The water-saturated group is to put the coal specimen into the vacuum bottle with good sealing performance, fill the vacuum bottle with water until the water completely covers the coal samples, and then install the sealing piston for vacuuming. The coal specimens were removed and weighed every 6h until the weight of the samples no longer changed, then the coal specimens had reached a water-saturated state;
Thanks very much for your serious attitude and professional advice. All modified parts of the revised manuscript have been marked in red.
Comment 11: Section 2.1, Para 2 "The experimental group was to place the coal samples completed with full water" does not make sense. Please rewrite.
Response:
Thank the reviewers for their professional suggestions. The authors have revised the manuscript for inappropriate expressions:
(Section 2.1, line 100) The experimental group consisted of subjecting water-saturated coal specimens to a temperature-controlled chamber set at -18℃ for a duration of 24 h.
Thanks very much for your serious attitude and professional advice. All modified parts of the revised manuscript have been marked in red.
Comment 12: Figure 1: Please annotate the diameter and height of the cylinder in the figure itself, so it is easy for the readers to judge the scale of samples.
Response:
Many thanks for this comment. Regarding this opinion of the reviewer, we have made the following supplements and improvements:
Figure 1. Standard coal specimen for testing
Thanks very much for your serious attitude and professional advice. All modified parts of the revised manuscript have been marked in red.
Comment 13: Section 2.2, Para 1, line 1 "In this study, fracturing and seepage experimental system for multi-physical field and multiphase coupling of porous media (THM-2) developed by Chongqing University, as shown in Fig. 2." does not make sense. Did you forget to say "was used" at the end?
Response:
Thank the reviewers for their professional suggestions. The authors have revised the manuscript for inappropriate expressions:
(Section 2.2, line 110) In this study, fracturing and seepage experimental system for multi-physical field and multiphase coupling of porous media (THM-2) developed by Chongqing University was used, as shown in Fig. 2.
Thanks very much for your serious attitude and professional advice. All modified parts of the revised manuscript have been marked in red.
Comment 14: Section 2.2, Para 1, line 2 "measurement and control system, etc." please do not write "etc." without giving either a reference to the complete equipment specifications, or writing the actual complete list explicitly.
Response:
Thank the reviewers for their professional suggestions. The authors have revised the manuscript for inappropriate expressions:
(Section 2.2, line 112) The system consists of a mainframe, electro-hydraulic servo hydraulic pump station, pneumatic and hydraulic pressure supply system, measurement and control system.
Thanks very much for your serious attitude and professional advice. All modified parts of the revised manuscript have been marked in red.
Comment 15: Section 2.2, Para 1, line 5 "The system can simulate porous media". Did you mean "The system can test (or perform experiments on) porous media"?
Response:
Thank the reviewers for their professional suggestions. The authors have revised the manuscript for inappropriate expressions:
(Section 2.2, line 118) The system is capable of simulating multiple coupled conditions such as in-situ stress, mining-induced stress, temperature, fluid pressure, and conducting uniaxial, triaxial mechanical tests, hydraulic fracturing tests, permeability tests, etc., on coal, sandstone, and other materials. It possesses high reliability and accuracy.
Thanks very much for your serious attitude and professional advice. All modified parts of the revised manuscript have been marked in red.
We tried our best to improve the manuscript and made some changes to the manuscript. These changes will not influence the content and framework of the paper. And here we did not list the changes but marked in red in the revised manuscript. We appreciate for Reviewers' warm work earnestly and hope that the correction will meet with approval.
Reviewer 2 Report
1. How representative are the tested coal specimens of real-world coal seams with low porosity, low permeability, and high inhomogeneity?
2. Can you provide more information about the freezing and thawing process used in the study? Were the freezing and thawing cycles representative of natural conditions?
3. How well do the triaxial permeability test conditions reflect the actual geological conditions encountered in coal mining operations?
4. What are the potential limitations or biases in using the Darcy's law expression to calculate permeability in this study?
5. Were there any significant variations observed in the mechanical response characteristics and permeability evolution of different coal samples from different locations or depths?
6. How do the observed stress-strain curves and mechanical response characteristics of coal specimens in different states compare to previous studies on similar materials?
7. Can you explain the reasons behind the observed differences in strength and permeability among dry, saturated, and freeze-thawed coal specimens?
8. How do the obtained permeability values for coal specimens under different confining pressures compare to the permeability values reported in other studies?
9. Were any additional tests or analyses conducted to confirm the accuracy and reliability of the permeability measurements?
10. What is the significance of the brief increases in permeability observed during the loading process? Do they have any practical implications for coal mine gas extraction?
11. Can you provide more insight into the mechanisms responsible for the observed permeability evolution patterns during the loading process?
12. What is the physical basis for the observed differences in permeability evolution between dry coal, saturated coal, and freeze-thawed coal specimens?
13. How do the observed strain variation rates and stress variation rates of coal specimens correlate with the changes in permeability?
14. What are the practical implications of the study's findings for coal mining operations in terms of safe and efficient gas extraction from low-permeability coal seams?
15. Are there any recommendations or suggestions for further research based on the findings and limitations of this study?
The quality of English language in the manuscript is generally good. The writing is clear and concise, and the technical terminology is used appropriately. However, there are a few areas where grammar and sentence structure can be improved to enhance readability and ensure a smoother flow of information.
Author Response
Reviewer #2:
Comments and Suggestions for Authors
Comment 1: How representative are the tested coal specimens of real-world coal seams with low porosity, low permeability, and high inhomogeneity?
Response:
Many thanks for this professional comment, regarding this question of the reviewer, we give the following explanation:
The coal specimens used in this paper were taken from the real coal seam of Baijiao coal mine of Sichuan Coal Group Furong Company, Gongxian County, Yibin City, Southwest China. It can be seen from Fig. 1 that the coal specimens have obvious laminations, in addition, according to the test results (Fig 4), the permeability of the coal specimens is extremely low, which is in line with the general characteristics of low porosity, low permeability, and high heterogeneity.
Comment 2: Can you provide more information about the freezing and thawing process used in the study? Were the freezing and thawing cycles representative of natural conditions?
Response:
Many thanks for this professional comment, regarding this question of the reviewer, we give the following explanation and modification:
The authors add information about the freezing and thawing process in the latest manuscript.
(Section 2.1, line 102)The frozen coal specimens were then thawed in room temperature water (approximately 18°C) for 24h. This cycle was repeated 5 times.
In natural conditions, cyclic freeze-thaw phenomena may indeed occur. However, the repeated freeze-thaw of coal seams under natural conditions is not the focus of this study. This study focuses on the impact of freezing-induced frost heave forces caused by ice formation on coal permeability and mechanical properties during the freeze-thaw process. The significance of this study lies in elucidating the mechanism of permeability increase by freeze-thaw.
Thanks very much for your serious attitude and professional advice. All modified parts of the revised manuscript have been marked in red.
Comment 3: How well do the triaxial permeability test conditions reflect the actual geological conditions encountered in coal mining operations?
Response:
Many thanks for this professional comment, we give the following explanation:
Triaxial permeability testing, as a commonly used technique, can to some extent reflect the conditions of the actual formation. This study mainly considers the triaxial stress and gas pressure of coal in real formation, which are important factors in coal mine gas disasters. However, there are still some factors that have not been taken into account in this study compared to the real formation. We will carefully consider the opinions of the reviewers and incorporate them into future work.
Comment 4: What are the potential limitations or biases in using the Darcy's law expression to calculate permeability in this study?
Response:
Many thanks for this professional comment, we give the following explanation:
Darcy's law has three main limitations:
- Darcy's law is based on the assumption of linear flow, where the permeability is assumed to vary linearly with the pressure gradient. However, in actual rocks, permeability characteristics can be influenced by nonlinear effects. For example, the permeability may change with the variation in flow velocity, limiting the applicability of Darcy's law.
- Darcy's law assumes that the rock permeability is isotropic, meaning the permeability is equal in all directions. However, in reality, rocks often exhibit anisotropic permeability, where the permeability varies in different directions.
- Darcy's law only considers the overall permeability of rocks and neglects the complex pore structure and flow paths within the rock. The permeability of real rocks is influenced by factors such as pore structure, connectivity, and porosity, which are not adequately considered in Darcy's law.
Comment 5: Were there any significant variations observed in the mechanical response characteristics and permeability evolution of different coal samples from different locations or depths?
Response:
Many thanks for this professional comment, we give the following explanation:
The coal specimens used in this study were taken from the same location, at the same depth, and from the same large coal block. Therefore, this study did not consider the mechanical and permeability characteristics of coal samples collected from different locations or depths. However, the questions raised by the reviewers are worthy of consideration and further research. The authors acknowledge that coal from different locations or depths can have variations in its physical and mechanical properties, including coal rank, moisture content, pore structure, and gas content, which can result in significant differences in their mechanical response and permeability evolution.
Comment 6: How do the observed stress-strain curves and mechanical response characteristics of coal specimens in different states compare to previous studies on similar materials?
Response:
Many thanks for this professional comment, we give the following explanation:
For different types of natural rock materials, there are variations in mineral composition, lithification type, pore structure, and damage state. In this study, differences were observed in the stress-strain curves and mechanical response characteristics of coal samples under different conditions compared to previous research. These differences mainly manifest in terms of strength and strain, but the overall trends are similar.
Comment 7: Can you explain the reasons behind the observed differences in strength and permeability among dry, saturated, and freeze-thawed coal specimens?
Response:
Many thanks for this professional comment, we give the following explanation and modification:
Dry coal specimens typically have a more compact structure, which helps to improve their strength. Saturated coal specimens contain pore water, which results in the involvement of water in the load transfer process when pressure is applied. The presence of water reduces the effective stress transfer between rock particles, as water has relatively high compressibility and flowability, resulting in a decrease in the overall strength of saturated rock samples. Additionally, the mineral composition of coal can also affect its strength, as some minerals may cause a decrease in the strength of coal samples due to hydrolysis. In a low-temperature environment, pore water freezes and expands during freezing. This leads to stress concentration and crack propagation within the specimen. When such cracks occur repeatedly due to freeze-thaw cycles, they further contribute to a decrease in the strength and eventual failure of the specimen.
Dry coal samples typically have lower moisture content, which can enhance their permeability. When a sample becomes saturated, water fills the pores of the coal, increasing the moisture content within the sample, which may affect gas flow and result in decreased permeability of the coal sample. During the freeze-thaw process, water undergoes expansion and contraction, potentially leading to the destruction of coal samples, enlargement of pores, and increased connectivity, ultimately increasing permeability. Furthermore, the mineral composition of coal can also affect its permeability. Different types of mineral components exhibit different physical and chemical properties, which may have varying effects on coal samples. Certain minerals may undergo physical or chemical changes during saturation and freeze-thaw processes, thereby increasing the permeability of coal samples.
Comment 8: How do the obtained permeability values for coal specimens under different confining pressures compare to the permeability values reported in other studies?
Response:
Many thanks for this professional comment, we give the following explanation:
Compared to other studies, the permeability of the coal specimens in this study is lower at different confining pressures (the permeability of the coal in this study is in the order of 10-17m2, while many of the other studies of the coal are in the order of 10-15 m2), and the permeability even decreases to zero under the action of axial stress.
Comment 9: Were any additional tests or analyses conducted to confirm the accuracy and reliability of the permeability measurements?
Response:
Many thanks for this professional comment, we give the following explanation and modification:
(Section 2.2, line 122) Before conducting the tests, we perform a seal integrity check on the testing equipment to ensure that our permeability system has good sealing performance prior to formal testing.
Thanks very much for your serious attitude and professional advice. All modified parts of the revised manuscript have been marked in red.
Comment 10: What is the significance of the brief increases in permeability observed during the loading process? Do they have any practical implications for coal mine gas extraction?
Response:
Many thanks for this professional comment, we give the following explanation:
The temporary increase in permeability during loading indicates the optimization of pore connectivity, which is a crucial phenomenon reflecting the state of coal's pore-fracture structure. In practical coal mining operations, this could potentially serve as a precursor to coal and gas outburst disasters. Understanding the mechanisms behind this phenomenon is of significant importance in reducing coal mine gas disasters and optimizing gas extraction processes.
Comment 11: Can you provide more insight into the mechanisms responsible for the observed permeability evolution patterns during the loading process?
Response:
Many thanks for this professional comment, we give the following explanation and modification:
(Section 4.2, line 333) Those disconnected closed pores and semi-closed pores become potential gas flow channels, the permeability of the coal specimen is relatively higher.
(Section 4.2, line 351) During the freeze-thaw process, water undergoes expansion and contraction, potentially leading to the destruction of coal samples, enlargement of pores, and increased connectivity, ultimately increasing permeability. Furthermore, the mineral composition of coal can also affect its permeability. Different types of mineral components exhibit different physical and chemical properties, which may have varying effects on coal samples. Certain minerals may undergo physical or chemical changes during saturation and freeze-thaw processes, thereby increasing the permeability of coal samples.
Thanks very much for your serious attitude and professional advice. All modified parts of the revised manuscript have been marked in red.
Comment 12: What is the physical basis for the observed differences in permeability evolution between dry coal, saturated coal, and freeze-thawed coal specimens?
Response:
Many thanks for this professional comment, we give the following explanation:
The physical basis for the differences in permeability evolution between dry coal, saturated coal, and freeze-thaw coal lies in the mineral composition, pore structure, water effects, and the action of freeze expansion, as follows:
Different types of mineral components in coal have different properties, which can have varying effects on the deformation of coal samples. Some minerals may undergo physical or chemical changes during saturation and freeze-thaw processes, resulting in changes in the permeability of coal samples. Compared to dry coal samples, when a sample is saturated, water fills the pores of the coal, increasing the moisture content within the sample, which may lead to a decrease in permeability. The presence of water increases the pore water pressure, thereby affecting the stress state within the rock and indirectly impacting the permeability of coal specimens. Freeze expansion can cause stress concentration and crack propagation within rock samples. When freeze-thaw cycles occur repeatedly, such cracks further develop, leading to an increase in permeability. During the freeze-thaw process, water undergoes expansion and contraction, potentially causing the destruction of coal samples and enlargement of pores.
Comment 13: How do the observed strain variation rates and stress variation rates of coal specimens correlate with the changes in permeability?
Response:
Many thanks for this professional comment, we give the following explanation:
During the testing process, a displacement loading method was used in the axial direction, and the axial stress variation rate can reflect the stress and damage state of the coal specimen. The radial strain variation rate can reflect the rate of crack propagation. These parameters can help explain the reasons for the evolution of permeability.
Comment 14: What are the practical implications of the study's findings for coal mining operations in terms of safe and efficient gas extraction from low-permeability coal seams?
Response:
Many thanks for this professional comment, we give the following explanation:
The results of this study have important practical significance in coal mining. The extraction of low-permeability coal seams and their associated gas is a complex process that requires consideration of factors such as permeability, physical and mechanical properties of the coal seam, gas content, and pressure. Conducting laboratory tests to obtain key physical and chemical property data can help evaluate the feasibility and production potential of coal seams and gas extraction.
Firstly, laboratory tests can assist in determining the permeability of the coal seam, which refers to the ability of natural gas to flow through the coal seam. This is crucial in determining the feasibility of coal seam extraction, as low-permeability coal seams require special extraction techniques and processes. Secondly, the research results can also analyze the characteristics of gas migration within the coal seam. This data can help mining engineers assess the recoverable reserves in the coal seam and develop reasonable extraction plans. Additionally, it provides a reference for the recovery and utilization of natural gas during the mining process. Furthermore, the study results indicate that the permeability evolution of coal after freeze-thaw still conforms to existing classical models, providing a basis for further research in this area.
Comment 15: Are there any recommendations or suggestions for further research based on the findings and limitations of this study?
Response:
Many thanks for this professional comment, we give the following explanation:
This study has revealed the evolution patterns of coal permeability under different conditions (dry, saturated, freeze-thaw). Furthermore, this study has provided preliminary insights into the mechanisms influencing the permeability of low-permeability coal layers due to water and freeze-thaw effects. However, the resolution of this issue is still under investigation. In our future work, we will focus on developing methods and technologies to address the challenge of reducing the impact of water and fractured coal particles on coal permeability.
Reviewer 3 Report
1. In abstract: the authors mention that the triaxial mechanics and percolation laws of dry, saturated, and freeze-thaw coal rocks were obtained, and the mechanism of the influence of freeze-thaw on the permeability evolution law of coal was revealed. It is hard to understand what the significant effect and the mechanism are by reading the abstract. It would be better to use data to support your statement.
2. Part 2.1: what is the basis for placing water-saturated coal at -18 degrees Celsius? What is its engineering significance?
3. Part 2.3: why do you choose CO2 for injection? In the context of coalbed methane extraction and coal mine gas safety, does it make sense for engineering?
4. Part 3.1: Variable symbols in the manuscript need to be italicized.
5. Part 3.3: Eq. (3) is a very classic model; however, it cannot consider the complexity of true coal. There are other modified models which can include multiple influencing factors to compute porosity. Why not use them?
6. Part 4.2: which numerical software that you used? This should be explained in the text.

Minor editing of English language required
Author Response
Reviewer 3:
Comment 1: In abstract: the authors mention that the triaxial mechanics and percolation laws of dry, saturated, and freeze-thaw coal rocks were obtained, and the mechanism of the influence of freeze-thaw on the permeability evolution law of coal was revealed. It is hard to understand what the significant effect and the mechanism are by reading the abstract. It would be better to use data to support your statement.
Response:
Many thanks for this professional comment, we give the following explanation and modification:
(Abstract, line 17) The triaxial mechanics and percolation laws of dry, saturated and freeze-thaw coal rocks were obtained, the results show that saturated coal have the lowest initial permeability and freeze-thawed coals have the highest initial permeability. By analyzing the effects produced by water and freezing and thawing on coal specimens, and the mechanism of the influence of freeze-thaw on the permeability evolution law of coal was revealed.
Thanks very much for your serious attitude and professional advice. All modified parts of the revised manuscript have been marked in red.
Comment 2: Part 2.1: what is the basis for placing water-saturated coal at -18 degrees Celsius? What is its engineering significance?
Response:
Many thanks for this professional comment, we give the following explanation:
The choice of -18°C is mainly based on two considerations:
- It is relatively easy to achieve a constant temperature environment of -18°C, and it can ensure that the water freezes.
- It excludes the influence of thermal stress caused by excessively low temperatures.
In engineering, natural conditions of -18°C do exist, and it can provide experimental data and theoretical support for low-temperature liquid injection increases permeability.
Comment 3: Part 2.3: why do you choose CO2 for injection? In the context of coalbed methane extraction and coal mine gas safety, does it make sense for engineering?
Response:
Many thanks for this comment. Regarding this opinion of the reviewer, we give the following explanation:
CO2 gas injection was chosen because it has some physical properties similar to CH4 (the dynamic viscosity of CH4 is 1.1067× 10-5pa·s, and that of CO2 is 1.4932× 10-5pa·s, they differ very little and do not have a big effect on permeability). There are huge safety risks associated with prolonged injection of CH4 gas, because it is a kind of flammable and explosive gas. There are many circuit systems and high pressure and high temperature oil pumps in the laboratory, which may cause CH4 gas to explode. In order to ensure the safety of the experiment, CO2 was selected to replace CH4.
Liquid carbon dioxide requires a high pressure and a suitable temperature (the critical pressure is 7-8 MPa, the critical temperature is 31.1℃, and in an environment of 25℃, the pressure is at least 6.5MPa), which requires high test requirements. So it is difficult to ensure that carbon dioxide in the sample is always kept in the liquid state. Flow monitoring is also more difficult, so gaseous carbon dioxide is used.
The reason why liquid injection is not used, because the resistance between the liquid and the rock must be overcome in the process of seepage. Unlike gas, when liquid percolates through pores, the flow rate is higher near the pore center and lower close to the pore wall. However, gas does not have this phenomenon, the velocity of gas molecules on the pore wall is basically the same as that in the pore center, which is called Klinkenberg effect. If the Klinkenberg effect is ignored, this will lead to a decrease in permeability and make the measurement of permeability more difficult. CO2 gas injection is a relatively convenient, safe and scientific method.
Thanks very much for your serious attitude and professional advice. All modified parts of the revised manuscript have been marked in red.
Comment 4: Part 3.1: Variable symbols in the manuscript need to be italicized.
Response:
Many thanks for this comment. Regarding this opinion of the reviewer. And it was revised in the manuscript.
The authors have updated the variable symbols in the latest manuscript to be italicized.
(Section 3.2, line 191) where k is the effective permeability, m2; Q is the gas seepage flow, m3/s; P0 is the standard atmospheric pressure (about 0.1 MPa), MPa; μ is the CO2 gas dynamic viscosity, 1.4932×10−11 MPa s; L is the length of the specimen, m; P1 is the inlet gas pressure (1.5 MPa), MPa; P2 is the outlet gas pressure (0.1 MPa), MPa; A is the effective area of gas seepage, m2.
Thanks very much for your serious attitude and professional advice. All modified parts of the revised manuscript have been marked in red.
Comment 5: Part 3.3: Eq. (3) is a very classic model; however, it cannot consider the complexity of true coal. There are other modified models which can include multiple influencing factors to compute porosity. Why not use them?
Response:
Many thanks for this comment. Regarding this opinion of the reviewer, we give the following explanation:
As mentioned by the reviewer, the complexity of real coal was not considered in the model. In Section 4.2, our main focus was on discussing and validating the model’s fit with the test data. Although Eq. (3) did not take into account the influence of rock complexity on permeability, based on the results, we found that the model showed a better match with the experimental data as indicated by the goodness of fit exceeding 0.99. This suggests that we have found a permeability model that aligns well with coal after freeze-thaw cycles. Limited by the length of this paper, we do ignore some classic revision models, which are not reflected in this paper, but this will become the focus of our future research.
Comment 6: Part 4.2: which numerical software that you used? This should be explained in the text.
Response:
Many thanks for this comment. Regarding this opinion of the reviewer, we give the following explanation:
The numerical analysis software MATLAB (matrix & laboratory) was used for all fitting calculations in this study.
Round 2
Reviewer 2 Report
No Comments
Minor editing of English language required